# Losing Weight after Menopause with Minimal Aerobic Training and Mediterranean Diet

**DOI:** 10.3390/nu12082471

**Published:** 2020-08-17

**Authors:** Mauro Lombardo, Marco Alfonso Perrone, Elena Guseva, Giovanni Aulisa, Elvira Padua, Chiara Bellia, David Della-Morte, Ferdinando Iellamo, Massimiliano Caprio, Alfonso Bellia

**Affiliations:** 1Department of Human Sciences and Promotion of the Quality of Life, San Raffaele Roma Open University, 00166 Rome, Italy; elena.guseva@uniroma5.it (E.G.); giovanni.aulisa@uniroma5.it (G.A.); elvira.padua@uniroma5.it (E.P.); david.dellamorte@uniroma2.it (D.D.-M.); massimiliano.caprio@uniroma5.it (M.C.); bellia@med.uniroma2.it (A.B.); 2Department of Systems Medicine, University of Rome “Tor Vergata”, 00133 Rome, Italy; iellamo@uniroma2.it; 3Division of Cardiology, University of Rome Tor Vergata, 00133 Rome, Italy; marco.perrone@uniroma2.it; 4School of Human Movement Science, University of Rome “Tor Vergata”, 00133 Rome, Italy; 5Institute of Clinical Biochemistry, Clinical Molecular Medicine and Laboratory Medicine, Department of Biomedicine, Neuroscience and Advanced Diagnostics, University of Palermo, 90127 Palermo, Italy; chiara.bellia@unipa.it; 6The Evelyn F. McKnight Brain Institute, Department of Neurology, Miller School of Medicine, University of Miami, Miami, FL 33136, USA; 7Laboratory of Cardiovascular Endocrinology, IRCCS San Raffaele Pisana, 00166 Rome, Italy

**Keywords:** Mediterranean diet, body composition, menopause, weight loss, chronic degenerative diseases

## Abstract

Objective: It is a common belief that menopausal women have greater difficulty losing weight. The aim of this study was to assess the efficacy of a Mediterranean diet (MD) to promote weight loss in postmenopausal women. All participants were prescribed a hypocaloric traditional MD, tailored to the individual. Subjects were asked not to begin any kind of physical activity. Body composition was measured at the beginning and after 8 weeks of treatment. In total, 89 women (age 52.8 ± 4.5 years, BMI 30.0 ± 5.2 kg/m^2^, fat mass 31.6 ± 10.5 kg) were divided into two groups: the first group consisted of fertile women over 45 years of age, the second group consisted of those diagnosed as menopausal. All women had an improvement in body composition (fat mass −2.3 ± 2.1 kg, *p* < 0.001; protein −0.1 ± 0.7 kg, *p* = 0.190) and blood pressure values. No differences were found between the two groups except for a higher reduction of low-density lipoprotein in the menopausal group (*p* = 0.035). A positive significant correlation between plant to animal protein ratio and fat-free mass variation was found in the menopausal group. These data suggest that a high adherence to a traditional MD would enable menopausal women to lose fat mass and maintain muscle mass with no significant difference to younger women. Fat mass reduction provides menopausal women with improved cardiovascular and metabolic risk factors.

## 1. Introduction

Menopause is a period in a woman’s life when weight maintenance may be particularly difficult [1]. Obesity is more common in men than in women before 45 years of age, but, after this point, the trend reverses [2]. It is estimated that women gain an average of 1–2 kg during the perimenopausal transition [3,4]. During the menopause, there is a shift in fat distribution and storage from the hips to the waist. There are a number of hormonal changes, such as decreased levels of estrogen and increased levels of circulating androgens, that lead to weight gain, particularly in terms of visceral fat [5]. In consequence, visceral fat mass increases by 44% during menopause, and the mass of gynoid fat rises by approximately 32% [6].

In addition to hormonal variations, it is relevant to consider the contribution of environmental factors such as reduced physical activity, Westernized eating patterns and common emotional eating disorders related to psychological discomfort [7]. Thus, various lipid metabolic disorders may lead to the development of metabolic syndromes including cardiovascular diseases and type 2 diabetes [8]. For this reason, metabolic syndrome is found in menopausal women three times more often than before the menopause [5]. This increased risk seems to be caused more by changes during the menopausal transition than by postmenopause [9].

Proper nutrition plays a fundamental role in the prevention of chronic degenerative diseases during the menopause [10]. In a study of women aged 50 to 60 years, fruit, vegetables and wholemeal bread were the most frequently consumed products in the healthy diet group. Significant correlations were observed with mental health, functioning in society, emotionality, vitality and well-being [11]. However, modest associations were observed for some single nutrients or food items with ovarian reserve and age at menopause [12]. Dietary intervention plus exercise provided better body weight and body composition (BC) parameters than calorie-restricted diet interventions alone in overweight and obese peri- and postmenopausal women [13].

Few studies have compared the efficacy of diet therapies in promoting weight loss in postmenopausal women [14]. The aim of this study is to assess whether menopausal status affects weight loss and BC changes in a homogeneous group of women over 45 years of age undergoing a traditional Mediterranean-type low-calorie diet.

## 2. Materials and Methods

### 2.1. Subjects

Subjects were selected from those who attended a nutrition clinic in Rome, Italy for the first time during the last trimester of 2019. All participants referred to the center were motivated to lose weight and were not informed of the different study groups within the study once they had agreed to take part. The criteria for eligibility were an age of 45 or more, a body mass index (BMI, the weight in kilograms divided by the square of the height in meters) of at least 25 kg·m^−2^ but less than 45 kg·m^−2^, a stable body weight over the 2 months before the baseline, and no change in medication for at least 3 months. Exclusion criteria were as follows: pregnancy or nursing; diabetes mellitus; chronic kidney disease; glucocorticoids, estrogens and anti-convulsant therapies; history of cardiovascular, neoplastic or other systemic diseases (both chronic and acute); any medication use known to affect weight/energy expenditure; alcohol intake (more than 3 drinks/day). The study was approved by the Ethics Committee of the University Hospital “Tor Vergata” (ID number 41.17) and all subjects gave informed consent. The study was conducted in accordance with the Declaration of Helsinki.

### 2.2. First Visit

All participants underwent a comprehensive medical evaluation including a clinical history, physical examination and anthropometric parameter measurement with BC assessment, blood pressure measurement and a blood sample. Blood pressure was measured after at least 10 min of rest in the sitting position, with a standard mercury sphygmomanometer on both arms. For all factors, mean data were determined from two measurements, and mean results were reported.

### 2.3. Body Composition Assessment

Weight and height were measured after an overnight fast. During the measurements, the subjects were allowed to wear only underwear. BMI was calculated as weight (kg) divided by the square of the height (m^2^). A standard cloth tape measure was used to record waist (WC) and hip (HC) circumferences. WC was taken at the midpoint between the lateral iliac crest and the lowest rib. HC was measured at the widest portion of the buttocks. BC was recorded with a BIA Tanita BC-420 MA, a validated instrument that uses dual-X-ray absorptiometry (DXA) [15] and measures values from a standing position and without the use of electrodes, to within 100 g. Patients were required to observe the following guidelines before BC evaluation: testing to be performed 3 h or more after meals and at least 3 h after waking up; no alcohol to be consumed and no severe restriction of caloric intake within 48 h of testing; no diuretics to be used within 7 days of testing and 12 h or more after hard, high intensity physical activity; urination just before testing; testing to avoid the menstruation period. Fat mass (FM), fat-free mass (FFM), protein and hydration status (total body water, TBW) were considered.

### 2.4. Dietary Intervention

All participants were placed on a two-month hypocaloric nutritionally balanced diet tailored to the individual. The moderate-fat, restricted-calorie Mediterranean diet (MD) was elaborated using nutritional software (Winfood 2.8; Medimatica Srl, Martinsicuro, Italy).

The main features of the prescribed diet were similar to the traditional Italian Nicotera’s MD [16]. This is a diet rich in plant-based foods such as fruits and vegetables, legumes and whole grains. The various components, expressed as percentages of the total calorie intake, were cereals 50–59%, extra virgin olive oil 13–17%, vegetables 2.2–3.6%, potatoes 2.3–3.6%, legumes 3–6%, fruit 2.6–3.6%, fish 1.6–2%, red wine 1–6%, meat 2.6–5%, dairy products 2–4% and very low amounts of eggs and animal fats. Patients were asked to use herbs and spices instead of salt to flavor foods, to reduce red meat to no more than twice a week and to eat fish at least twice a week. The main sources of daily added fat were 25 to 35 g of olive oil and a handful of nuts (20 g) per day.

The recommended composition of the dietary regimen was as follows: carbohydrates 50–60%, proteins 10–15% (of which more than 50% should be comprised of vegetable proteins), total fat less than 30% (saturated fat less than 7% and cholesterol consumption less than 300 mg per day), and more than 30 g of fiber. Diets were divided into three main meals and two snacks, with a 500–600 kcal/day deficit compared to the estimated total energy expenditure TEE [17]. The after-dinner snack was prescribed only when requested.

It was recommended that patients only performed minimal aerobic training.

### 2.5. Dietary Intake

Patients were required to record their dietary intake (consecutively over two weekdays and one weekend day) in a semi-quantitative weighed food record at baseline, and then weekly throughout the follow-up. To avoid under-reporting, all subjects whose reported intake was <110% of their estimated basal metabolic rate were excluded, as explained by Mendez et al. [18]. Subjects were required to weigh and measure their food using home digital scales. Dietary composition was examined using professional software (Nutritics, Dublin, Ireland, 2019).

### 2.6. Follow-Up Visits

The same parameters described above were measured after 4 and 8 weeks. Results after 8 weeks were compared with those of the baseline tests. This involved individual sessions (anthropometric, BC, evaluation of nutrient intake and diet adequacy). Thus, twice a month, patients were scheduled to meet the dietitian to reinforce the nutritional rehabilitation program that aimed to improve and promote changes in eating habits. Tables with nutritional values of foods were used to assist participants with grocery shopping. A web chat consultation service was also provided to families and the health care professionals caring for them, for patient support.

### 2.7. Laboratory Data

In a group of 30 patients, hematochemical tests were required. Blood samples were taken in the morning (between 7:00 AM and 9:00 AM). Total cholesterol, high-density lipoprotein (HDL) cholesterol and triglycerides were determined by automated conventional enzymatic methods. Low-density lipoprotein (LDL) cholesterol was calculated using Friedewald’s formula [19]. Serum levels of 25(OH)D and fasting insulin were determined by chemiluminescence. The hematochemical values were repeated at the end of the study, except for vitamin D which was only determined at the beginning.

### 2.8. Groups

The patients were separated into two groups. The first group (REGULAR) contained women over 45 years old with a regular menstrual cycle; the second group (MENOPAUSE) contained women in menopause. Menopause was defined as a woman’s FSH blood level being consistently elevated to 30 mIU/mL or higher and no menstrual period for a year.

### 2.9. Statistical Analysis

The data were analyzed with SOFA Statistics version 1.4.6 open source software. Results for descriptive statistics are expressed as mean ± standard deviation. All quantitative variables were tested for normality distribution using the Kolmogorov–Smirnov test. Dietary composition data (energy intake; percentage of carbohydrate, fat and protein; g saturated fat and fibre) were analyzed using means. Baseline differences in continuous variables between the groups’ subjects were assessed with Student’s *t*-test for unpaired data. Statistical comparisons of continuous variables among the groups were performed with independent sample *t*-tests. The relationship between age and menopause duration and BC parameter variations was evaluated. The Spearman’s coefficient was used to measure the correlation among variables and the corresponding *p*-values are reported for the significance of the correlation. For all analyses, a *p*-value < 0.05 based on a two-sided test was considered statistically significant.

## 3. Results

Baseline features of the 89 women recruited for the study (age 52.8 ± 4.5 years, 17 smokers) are shown in Table 1. One subject dropped out due to personal reasons. All subjects were white Europeans. Subjects had an average BMI of 30.0 ± 5.2 kg/m^2^, FM of 31.6 ± 10.5 kg, FFM of 45.0 ± 5.3 kg and TBW of 33.5 ± 4.6 kg. All participants were overweight or obese, with nearly 45% of the subjects recording a BMI over 30 and predominant abdominal obesity according to WC measurements (94.8 ± 13.6 cm). Hypertension was reported in 30 subjects. There were no significant differences between the two groups apart from the menopausal women having a significantly higher systolic blood pressure (SBP).

The composition of the dietary intake and the difference between groups are shown in Table 2. Analysis of the food diaries shows that the diets had a mean macronutrient composition of about 0.85 g/kg/day protein, 32% fat and 52% carbohydrates, with three main meals and two snacks spread throughout the day.

The differences for all subjects after 2 months are shown in Table 3. Analysis of the results of the entire sample, two months after the start of nutritional therapy, shows a significant difference in BMI of −0.8 ± 0.5 kg/m^2^, FM of −2.3 ± 2.1 kg and FFM of −0.7 ± 1.6 kg, of which TBW was −0.6 ± 1.1 kg and body protein was −0.1 ± 0.7 kg (*p* = 0.190). No significant correlation is observed between patients’ age and BC variation during treatment (Figure 1). The change in lifestyle also brought a significant improvement in body circumference and blood pressure. The variation in basal metabolic rate (BMR) was −25.6 ± 38.3 kcal/day. The weight loss allowed a significant improvement of some lipid blood values, such as total cholesterol and LDL, in the whole sample.

There were no significant differences between the two groups for any of these parameters, except for the variation of LDL (−28.2 ± 31.6 mg/dL vs. −7.7 ± 16.9 mg/dL; *p* = 0.035) where the reduction was greatest in the group of menopausal women.

Figure 2 shows the correlation between plant to animal protein ratio and FFM variation after lifestyle intervention. A positive significant correlation between these parameters was found in all subjects and in the menopause group (Figure 3).

The composition of the basal diets and the difference between groups are shown in Table 2. Analysis of the food diaries shows that the diets had a mean macronutrient composition of about 12% proteins (0.85 gr/kg/day), 32% fat and 56% carbohydrates. Throughout the day were three main meals and two snacks. The after-dinner snack was prescribed only for women requesting it.

The differences for all subjects after 2 months are shown in Table 3. Analysis of the results of the entire sample, two months after the start of nutritional therapy, shows a significant difference of BMI of −0.8 ± 0.5 kg/m^2^, FM of −2.3 ± 2.1 kg, FFM of −0.7 ± 1.6 kg of which TBW was −0.6 ± 1.1 and protein −0.1 ± 0.7 (*p* = 0.1862). The analysis of linear regression between patients’ age and BMI variation during treatment shows that dietary effectiveness tends to increase slightly with age (r = −0.123; *p* = 0.252, Figure 1). The change in lifestyle also brought a significant improvement in body circumference and blood pressure. The variation in basal metabolic rate (BMR) was −25.6 ± 38.3 kcal/die. The weight loss allowed a significant improvement of some lipid blood values, such as total cholesterol and LDL, in the whole sample.

There were no significant differences between the two groups for any of these parameters, except for the variation of LDL where the reduction was greatest in the group of menopausal women (−28.2 ± 31.6 mg/dL vs. −7.7 ± 16.9 mg/dL; *p* = 0.035).

## 4. Discussion

Overall, we found that high adherence to a traditional MD diet, with moderate protein and high fiber intake, had a similar effect on BC for free-living postmenopausal overweight/obese women and fertile women. Several studies have demonstrated that women undergoing perimenopause lose lean body mass and more than double their FM, and that this trend continues until 2 years after menopause when the BC stabilizes [3,6,7,20]. This phenomena is influenced more by reduced energy expenditure rather than increased energy intake [21]. Our data confirm other work which showed that a higher adherence to a healthy dietary pattern (the Mediterranean diet) is inversely associated with overweight/obesity in perimenopausal and postmenopausal women [22].

A cross-sectional study by Rolland et al. showed a decline of 0.6% per year in muscle mass after menopause [23]. The same study showed that, for an increase of 0.1 g/kg of daily protein intake, the drop in muscle mass was reduced by 0.62 kg. Our data revealed that a protein intake similar to the nutritional guidelines (0.82 g/kg) was enough to lose FM and maintain muscle mass in menopause patients. On the other hand, a recent review has found that intake exceeding the Recommended Daily Allowance (RDA) may be preferential in preserving muscle mass and function in ageing adults [24]. For instance, among post-menopausal women of 60–90 years, those in the low protein group had higher body fat and fat-to-lean ratio than those who consumed a higher protein diet [25].

Surprisingly, we found a correlation between a higher plant to animal protein ratio and a lower reduction in lean mass, both in the whole sample (Figure 2) and in the menopausal women group (Figure 3). Plant-based protein sources have a lower leucine content than animal-based protein sources. Furthermore, plant-based protein sources are classically considered as “deficient in certain essential amino acids” for body needs [26]. A previous study showed that higher plant protein intake, particularly from non-cereal products, with a proportional decrease in animal protein intake did not affect body weight maintenance or cardiometabolic risk factors [27]. A high protein content of plant origin, such as the traditional MD, would add further benefits to patients’ health. Other papers support the beneficial effect of a MD, which is rich in legumes, fiber, and monounsaturated fat, for weight loss [28,29]. This advantage would add to the other positive effects of consuming this type of protein in menopause, such as decreasing bone loss and risk of hip fracture [30], improving insulin sensibility [31,32] and decreasing cardiovascular disease risk [33].

Our data confirm the importance of the MD in menopausal patients for the improvement of cardiovascular and metabolic risk factors. Patients in the menopausal group had even greater improvement in LDL and triglyceride parameters. After the menopause, lipid metabolic disorders are common, mainly due to hormonal changes. It has been demonstrated that a high, but not medium, adherence to the MD is associated with a cardioprotective effect in peri and menopausal women [34]. A stronger adherence to a MD was associated with a lower lipid peroxidation (LPO) among healthy women [35]. LPO is actively involved in the inflammatory responses in atherosclerosis by interacting with immune cells (such as macrophages) and endothelial cells [36].

In this study, women lost a minimal, non-significant amount of muscle mass, although their physical activity was reduced and was only of the aerobic type. These findings may indicate that a hypocaloric MD, without prescription of physical activity, is sufficient to lose FM and preserve FFM in the short term. However, physical activity may be vital to modify the BC in menopausal women with obesity in the long term [37]. The slight reduction in resting metabolism shows that the MD may be an excellent option for maintaining weight loss in the long term [38]. Conflict exists as to whether perimenopausal weight gain is due to changes in energy metabolism related to aging or changes in sex hormones related to menopause. Several papers have demonstrated that aging is the primary driver of FM gain in menopausal women, rather than the constant change of hormone levels during menopause [39,40]. A reduction in BMR at menopause has been shown to be independent of changes in BC [41] and may be related to a reduction in the 24-h core temperature [42].

In our study, we found no correlation between age and improvement in BC (Figure 1). Patients had improvements regardless of their age and the age of menopause. In a previous study, the factors significantly related to ≥3% weight gain were weight change in the past 2 years, age at menopause, dietary fiber, fat, alcohol intake, and smoking [42].

Our study had several limitations. Firstly, the sample size was relatively small. Secondly, the methods for assessing the energy expenditure of BC should have been more precise than those used. Thirdly, the participants should have been studied for a longer period of time.

## 5. Conclusions

High adherence to the Mediterranean diet would ensure menopausal women lose fat mass and maintain muscle mass in the same way as younger women, without the prescription of “structured” physical activity. Increasing the plant protein content of a hypocaloric diet may lead to a preservation of muscle mass in menopausal women. Studies demonstrating long-term effects of diets on menopausal women are warranted.

## Figures and Tables

**Figure 1 nutrients-12-02471-f001:**
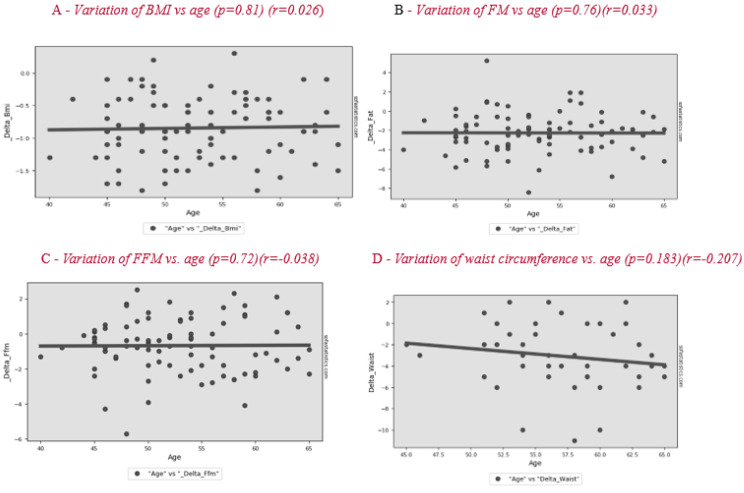
Correlation between age and variation of BMI (body mass index) (**A**), FM (fat mas) (**B**), FFM (fat-free mass) (**C**) and waist circumference (**D**) after the lifestyle intervention.

**Figure 2 nutrients-12-02471-f002:**
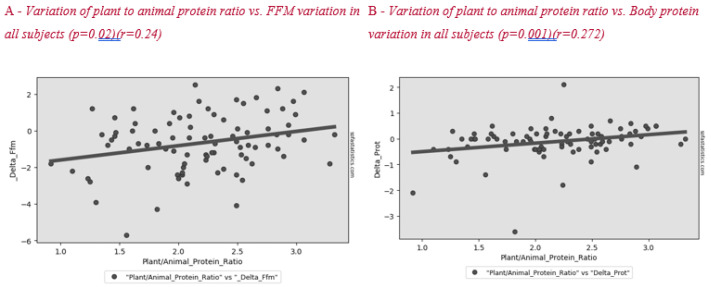
Correlation between plant to animal protein ratio and FFM (**A**) and body protein (**B**) variation in all subjects after the lifestyle intervention.

**Figure 3 nutrients-12-02471-f003:**
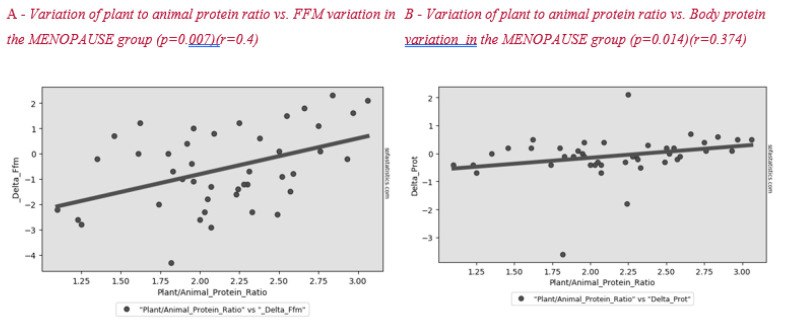
Correlation between plant to animal protein ratio and FFM (**A**) and body protein (**B**) variation in the MENOPAUSE group after the lifestyle intervention.

**Table 1 nutrients-12-02471-t001:** Body composition and other characteristics of study subjects.

**Variables**	**Menopause** **(*n*. 43)**	**Regular** **(*n*. 46)**	**Total** **(*n*. 89)**	**Between-Group** ***p*-Value**
Age	57.1 ± 4.9	48.8 ± 4.0	52.8 ± 4.5	<0.001
Smokers	11	6	17	
Height, m	162.4 ± 5.7	161.6 ± 6.4	162.0 ± 6.1	0.536
Weight, kg	80.8 ± 16.2	77.0 ± 14.1	78.8 ± 15.2	0.240
BMI, kg/m^2^	30.6 ± 5.4	29.5 ± 5.0	30.0 ± 5.2	0.321
Waist, cm	97.5 ± 15.3	92.2 ± 11.5	94.8 ± 13.6	0.067
Hip, cm	109.5 ± 9.8	108.2 ± 9.4	108.8 ± 9.6	0.525
BMR, kcal/day	1449.5 ± 190.0	1428.8 ± 179.0	1438.8 ± 183.6	0.598
SBP, mm Hg	133.8 ± 17.2	124.6 ± 13.3	129.0 ± 15.9	0.005
DBP, mm Hg	85.2 ± 13.0	80.6 ± 12.8	82.8 ± 13.0	0.096
Body Composition				
FM, kg	33.4 ± 11.1	29.9 ± 9.7	31.6 ± 10.5	0.116
FFM, kg	45.1 ± 5.4	44.9 ± 5.3	45.0 ± 5.3	0.860
TBW, kg	33.6 ± 4.8	33.4 ± 4.4	33.5 ± 4.6	0.838
Body protein, kg	11.5 ± 0.8	11.5 ± 1.0	11.5 ± 0.9	1.000
Haematochemical tests *				
	**Menopause** **(*n*. 15)**	**Regular** **(*n*. 15)**	**Total** **(*n*. 30)**	**Between-Group** ***p*-Value**
Glycaemia, mg/dL	92.7 ± 14.0	92.4 ± 9.4	92.6 ± 11.9	0.946
Total cholesterol, mg/dL	219.6 ± 41.7	202.9 ± 35.5	210.8 ± 39.2	0.248
LDL, mg/dL	143.6 ± 36.2	130.3 ± 29.4	136.9 ± 33.3	0.279
HDL, mg/dL	58.3 ± 12.6	58.9 ± 16.4	58.6 ± 14.7	0.911
Triglycerides, mg/dL	108.3 ± 65.2	92.0 ± 39.0	99.8 ± 53.4	0.413
Vitamin D3, ng/mL	22.4 ± 8.2	19.2 ± 12.4	20.9 ± 10.3	0.407

* Data of 30 patients. Data are expressed as mean values ± SD. Abbreviations—SD: standard deviation; *n*: number of subjects; ns: not significant; BMI: body mass index; LDL: low-density lipoprotein; HDL: high-density lipoprotein; FM: fat mass; FFM: fat-free mass; TBW: total body water; BMR: basal metabolic rate; SBP: systolic blood pressure; DBP: diastolic blood pressure.

**Table 2 nutrients-12-02471-t002:** Composition and % breakdown of meals of the basal dietary intake and differences between groups.

Variables	Menopause (*n*. 43)	Regular (*n*. 46)	Total (*n*. 89)	Between-Group *p*-Value
Calories, kcal	1456.3 ± 126.4	1453.5 ± 157.5	1454.9 ± 142.5	0.927
Carbohydrates, %	52.8 ± 3.1	51.9 ± 4.0	52.3 ± 3.6	0.257
Protein, %	17.5 ± 1.0	17.5 ± 1.2	17.5 ± 1.1	0.970
Lipids, %	32.2 ± 2.8	32.7 ± 3.4	32.5 ± 3.1	0.430
Carbohydrates, g	192.1 ± 17.9	188.7 ± 24.7	190.3 ± 21.6	0.462
Oligosaccharides, g	74.5 ± 9.2	71.3 ± 8.4	72.9 ± 8.9	0.090
Starch, g	98 ± 17.1	97 ± 17.1	97.5 ± 17	0.783
Total fiber, g	40.9 ± 3.3	41.3 ± 4.1	41.1 ± 3.7	0.615
Insoluble fiber, g	19.1 ± 1.7	19.2 ± 1.4	19.2 ± 1.5	0.762
Soluble fiber, g	6.7 ± 0.7	6.6 ± 0.8	6.7 ± 0.7	0.533
Protein, g	63.6 ± 5.9	63.3 ± 6.5	63.5 ± 6.2	0.821
Protein, g/kg	0.81 ± 0.14	0.84 ± 0.13	0.82 ± 0.13	0.257
Plant protein, g	36.9 ± 4.8	36.9 ± 3.8	36.9 ± 4.3	1.000
Animal protein, g	18.1 ± 4.8	17.3 ± 5.4	17.7 ± 5.1	0.463
Plant to animal protein ratio	2.1 ± 0.5	2.3 ± 0.8	2.2 ± 0.6	0.190
Lipids, g	52.2 ± 7.6	53 ± 8.7	52.6 ± 8.1	0.646
TUFA, g	35.7 ± 5.4	36.1 ± 6.1	35.9 ± 5.7	0.745
MUFA, g	26.9 ± 4	27.1 ± 4.7	27 ± 4.4	0.830
PUFA, g	8.8 ± 1.6	9 ± 1.8	8.9 ± 1.7	0.582
SFA, g	9.3 ± 1.6	9.2 ± 2	9.2 ± 1.8	0.796
Cholesterol, mg	77.9 ± 26.8	73.8 ± 29.3	75.7 ± 28.1	0.494
Alcohol, g	0	1.5 ± 6	0.8 ± 4.4	0.105
% BREAKDOWN AMONG MEALS (calories)				
Breakfast, %	18.3 ± 3	18.2 ± 3.6	18.2 ± 3.3	0.888
Morning Snack, %	9.2 ± 6.9	9.8 ± 8.6	9.5 ± 7.8	0.719
Lunch, %	30.5 ± 6.3	31.6 ± 8.4	31.1 ± 7.4	0.489
Afternoon Snack, %	12.3 ± 6.4	11.8 ± 6.8	12 ± 6.6	0.722
Dinner, %	28.9 ± 7.4	27.8 ± 7.4	28.3 ± 7.4	0.485
After Dinner, %	4.6 ± 3.3	5.6 ± 2.7	5.1 ± 3	0.120

Data are expressed as mean values ± SD. Abbreviations—*n*: number of subjects; TUFA: total unsaturated fatty acids; SFA: saturated fatty acids; MUFA: monounsaturated fatty acids; PUFA: polyunsaturated fatty acids.

**Table 3 nutrients-12-02471-t003:** Changes in parameters for study subjects between baseline and 2-month follow-up.

**Variables**	**TOTAL (*n*. 89)**	**Total Sample** ***p*-Value**	**MENOPAUSE (*n*. 43)**	**REGULAR (*n*. 46)**	**Between-Group** ***p*-Value**
△ Weight, kg	−3.4 ± 2.3	<0.001	−3.7 ± 2.6	−3.1 ± 1.9	0.215
△ BMI, kg/m^2^	−0.8 ± 0.5	<0.001	−0.9 ± 0.5	−0.8 ± 0.5	0.348
△ FM, kg	−2.3 ± 2.1	<0.001	−2.4 ± 2.0	−2.2 ± 2.1	0.647
△ FFM, kg	−0.7 ± 1.6	<0.001	−0.6 ± 1.6	−0.8 ± 1.6	0.557
△ TBW, kg	−0.6 ± 1.1	<0.001	−0.5 ± 1.1	−0.6 ± 1.2	0.684
△ Body protein, kg	−0.1 ± 0.7	0.190	−0.1 ± 0.8	−0.1 ± 0.5	1.000
△ Waist, cm	−3.1 ± 2.9	<0.001	−3.1 ± 3.1	−3.2 ± 2.8	0.873
△ Hip, cm	−2.8 ± 2.6	<0.001	−2.6 ± 2.9	−3.0 ± 2.3	0.471
△ BMR, Kcal	−25.6 ± 38.3	<0.001	−24.1 ± 40.2	−27 ± 36.9	0.724
△ SBP, mm Hg	−7.9 ± 15.2	<0.001	−9 ± 16.4	−6.8 ± 13.9	0.496
△ DBP, mm Hg	−5.1 ± 13.2	0.014	−7 ± 14.9	−3.3 ± 11.2	0.187
Haematochemical tests *					
	**TOTAL (*n*. 30)**	**Total Sample** ***p*-Value**	**MENOPAUSE (*n*. 15)**	**REGULAR (*n*. 15)**	**Between-Group** ***p*-Value**
△ Glycaemia, mg/dL	−2.3 ± 9.1	0.193	−2.5 ± 10.1	−2.2 ± 7.9	0.928
△ Triglycerides, mg/dL	−6.3 ± 48	0.493	−22.9 ± 56.2	8.2 ± 35.1	0.080
△ Total cholesterol, mg/dL	−15.3 ± 28.2	0.033	−22.8 ± 24.7	−9.3 ± 29.9	0.188
△ HDL, mg/dL	3.9 ± 10.5	0.059	6.0 ± 10.4	2.2 ± 10.6	0.330
△ LDL, mg/dL	−17.2 ± 26.4	0.028	−28.2 ± 31.6	−7.7 ± 16.9	0.035

Data are expressed as mean values ± SD. Abbreviations—*n*: number of subjects; BMI: body mass index; LDL: low-density lipoprotein; HDL: high-density lipoprotein; FM: fat mass; FFM: fat-free mass; TBW: total body water; BMR: basal metabolic rate; SBP: systolic blood pressure; DBP: diastolic blood pressure. * Data of 30 patients.

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
