# Peer review of "Losing Weight after Menopause with Minimal Aerobic Training and Mediterranean Diet"

_nutrients, 2020, doi:10.3390/nu12082471_

Round 1

Reviewer 1 Report

General:

  • the manuscript needs some English editing and revision overall
  • the title states "Losing weight after menopause with sport and mediterranean diet" but then in the methods section you state that "Patients were only recommended to perform minimal aerobic training" and the only intervention is diet. So I would suggest to change the title since it is misleading.
  • both "."and "," are used to indicate decimals. Please, homogenize througout the text and tables. 

Methods:

- you state that you recruited all overweight and obese women, but then the BMI specific criteria states BMI above 21, which is normal weight. I would suggest to clarify this issue. Also, the women you selected were the ones that went to an obesity center. Was this a voluntary appointment or was this a recommendation/redirection by other health professional? This is an important issue and depending on the willingness of women to go to this obesity, it might cause the sample to be biased.

- what does it mean the exclusion criteria "any lifestyle treatment in the year before"?

- follow-up visits: needs some clarification. At the beginning it states that the follow-up was done at 4 and 8 weeks, but later it states that participants had an appointment with a dietician 2 times/month measuring body composition, and other paramaters. Please clarify.

Results: 

- Table 1 shows a variable named "Protein, kg". What does it mean? It is not explained.

- Table 2: I think a better explanation is needed on how you got all the macronutrients data. In the methods section you mention some diet diaries, but it is not well explained. 

- Table 3: some of the p-values shown seem strange. e.g Triglycerides had a very high difference between groups but p-value=0.4931, while other 

You state"There were no significant differences between the two groups for any of these parameters, except 173 for the variation of LDL where the reduction was greatest in the group of menopausal women (-28.2 174 ± 31.6 mg/dL vs -7.7 ± 16.9 mg/dL; p=0.035)." but this p-value is not what is shown in table 3. Please revise. Also, a lot of the parameters shown in Table 3 show a p-value below 0.05, so the statement above does not seem to be correct. 

- In table 2 you show the macronutrients for the basal diet, but it would be good to include a similar table for the diet that proposed (the intervention), or, at least, discuss about the differences between baseline and proposed diets

Discussion:

- the first sentence is not clear and I suggest making a more clear statement in line with the results presented

- 2nd paragraph of discussion does not make sense in this context

- an improvement of the results presentation (see suggestions above) will have to follow an improvement of the discussion in line with the results.

Reviewer 2 Report

The present manuscript builds on a common belief that after menopause, l women have greater difficulty in losing weight. To address this question, they had 8 weeks of hypocaloric mediterranean diet for 89 fertile/menopausal women. The authors found that all women had improvement in body composition and blood pressure, whereas menopausal women had higher reduction of LDL.

Even though I find the topic of the study interesting, there are several concerns that I have listed below:

GENERAL COMMENS:

Unfortunately, I find both the introduction and the discussion shallow. Both sections need to be re-written with better focus and less opinions but concentrating on previous literature with appropriate references.

The paper needs to go through a language check, as there is a lot of spoken language in the text.

Throughout the tables, please use dot as a separator, not comma (now the authors have used both) and be consistent in how many decimal numbers you shown (now there is 3-4).

Throughout the text use the same font. Now it appears that there are several different font sizes that change even in the middle of a sentence.

DETAILED COMMENTS:

Lines 38-39: “Overweight and obese women gain more weight in the perimenopause period. The effects of low-calorie diets are often less effective for women than for men of the same age.” These kind of statements need a reference.

Lines 45-46: “Proper nutrition plays a fundamental role during the menopause, even more than in the rest of life, for the prevention of chronic degenerative disease.” How did the authors come to this conclusion? Please add references!

Lines 71-72: “Weight and height were determined while the subjects were fasting overnight and wearing only underwear.” I suppose the authors mean “Weight and height were measured after an overnight fasting. During the measuring the subjects were allowed to wear only underwear.”

Line 78: “..not eating or drinking too much..”. How was too much defined?

Lines 90-91: “All participants were placed on a two-month hypocaloric nutritionally balanced diet, tailored to the individual” Please explain how were the diets tailored.

Lines 189-190: “Our paper shows that weight loss remains steady with increasing age even in the absence of structured physical activity (Figure 1).” Were the authors studying the effects of aging or menopause?

Lines 191-192: “Our data also confirms the extraordinary usefulness of the MD in menopausal patients for the improvement of cardiovascular and metabolic risk factors.” I find that the authors declare quite strong statements such as the previous sentence without comparing other treatment methods either in their own study or from previous literature. It makes the text sound unprofessional, so please add more robust information in the discussion.

Tables 1 and 3: How is it possible that at the baseline the blood sample analysis (e.g. HDL and LDL) was measured from 15+15 subjects, but in Table 3 there are full N:s? (n=43-46).

Table 3: There is a clear trend in the change of triglycerides (p=0.080), please discuss this.

Round 2

Reviewer 1 Report

The paper has improved very much from the first version. Still needs some English editing, but minor. 

In general terms, I would re-phrase the conclusion,both in the abstract and conclusion sections. Now the abstract reads: "These data suggest that a high adherence to a traditional MD would enable menopausal women to lose fat mass and maintain muscle mass with no significant difference to younger women. Fat mass reduction provides menopausal women with improved cardiovascular and metabolic risk factors", that might lead to think that menopausal women have more benefits than non-menopausal women.

Abstract: I don't understand the sentence "A positive significant correlation between plant to animal protein ratio and fat-free mass variation was found in the menopausal group".

Review sentence 280-281: "High adherence to the Mediterranean diet would ensure menopausal women lose fat mass and maintain muscle mass without in the same way as younger women"

Author Response

REVIEWER N.1

The paper has improved very much from the first version. Still needs some English editing, but minor. 

We thank the reviewer for the useful remarks that allowed us to improve the quality of the paper

Point 1: In general terms, I would re-phrase the conclusion, both in the abstract and conclusion sections. Now the abstract reads: "These data suggest that a high adherence to a traditional MD would enable menopausal women to lose fat mass and maintain muscle mass with no significant difference to younger women. Fat mass reduction provides menopausal women with improved cardiovascular and metabolic risk factors", that might lead to think that menopausal women have more benefits than non-menopausal women.

Review sentence 280-281: "High adherence to the Mediterranean diet would ensure menopausal women lose fat mass and maintain muscle mass without in the same way as younger women"

Response 1: We rewrote the conclusions both in the abstract both in the main text to better focus on the results of our work.High adherence to the Mediterranean diet would ensure menopausal women to lose fat mass and maintain muscle mass without any significant difference with younger women. Positive effects of a high plant protein intake on lean body and muscle mass during diet-induced weight loss has been shown. Studies demonstrating long-term effects of diets on menopausal women are warranted”.

Point 2 Abstract: I don't understand the sentence "A positive significant correlation between plant to animal protein ratio and fat-free mass variation was found in the menopausal group".

Response 2: We rephrased the sentence. “Positive effects of a high plant protein intake on lean body and muscle mass during diet-induced weight loss has been shown.”

Reviewer 2 Report

Dear Authors,

Thank you for considering my suggestions and corrections, the manuscript has improved significantly.

There are still few minor changes I would recommend you to revise:

Abstract:

In the abstract you state that: "A positive significant correlation between plant to animal protein ratio and fat-free mass variation was found in the menopausal group" whereas in the discussion it is said that: (lines 240-242) "Surprisingly, we found a correlation between a higher plant to animal protein ratio and a lower reduction in lean mass, both in the whole sample (Figure 2) and in the menopausal women group".

Please be clear which statement is the truth and modify text accordingly. Also, use similar wording as you used in the discussion (e.g. "we found a correlation between a higher plant to animal protein ratio and a lower reduction in lean mass") and not less informative sentences such as "correlation between plant to animal protein ratio and fat-free mass variation was found", as fat-free mass variation does not tell the reader what you actually found.

Conclusions:

I would highly recommend the authors to re-write the Conclusions section. The word ensure is too strong in this context, and there is an extra "without" in the first sentence. I would also not use " " in scientific writing. It either is structures physical activity or not, it cannot be "kind of" structured.

Author Response

REVIEWER N.2

Thank you for considering my suggestions and corrections, the manuscript has improved significantly.

We thank the reviewer for the useful comments that allowed us to improve the quality of the paper.

Point 1: Abstract: In the abstract you state that: "A positive significant correlation between plant to animal protein ratio and fat-free mass variation was found in the menopausal group" whereas in the discussion it is said that: (lines 240-242) "Surprisingly, we found a correlation between a higher plant to animal protein ratio and a lower reduction in lean mass, both in the whole sample (Figure 2) and in the menopausal women group".

Response 1: We corrected the sentence in the abstract as you asked.

Point 2: Please be clear which statement is the truth and modify text accordingly. Also, use similar wording as you used in the discussion (e.g. "we found a correlation between a higher plant to animal protein ratio and a lower reduction in lean mass") and not less informative sentences such as "correlation between plant to animal protein ratio and fat-free mass variation was found", as fat-free mass variation does not tell the reader what you actually found. I would highly recommend the authors to re-write the Conclusions section. The word ensure is too strong in this context, and there is an extra "without" in the first sentence. I would also not use " " in scientific writing. It either is structures physical activity or not, it cannot be "kind of" structured.

Response 2: We rewrote the Conclusions section as requested.

It appears that high adherence to the Mediterranean diet helps menopausal women to lose fat mass and maintain muscle mass without any significant difference with younger women. Positive effects of a high plant protein intake on muscle mass during diet-induced weight loss have been shown. Studies demonstrating long-term effects of diets on menopausal women are warranted. “